# MAKING BATCH NORMALIZATION GREAT IN FEDERATED DEEP LEARNING

## ABSTRACT

Batch Normalization (BN) is commonly used in modern deep learning to improve stability and speed up convergence in centralized training. In federated learning (FL) with non-IID decentralized data, previous works observed that training with BN could hinder performance due to the mismatch of the BN statistics between training and testing. Group Normalization (GN) is thus more often used in FL as an alternative to BN. In this paper, we identify a more fundamental issue of BN in FL that makes BN inferior even with high-frequency communication between clients and servers. We then propose a frustratingly simple treatment, which significantly improves BN and makes it outperform GN across a wide range of FL settings. Along with this study, we also reveal an unreasonable behavior of BN in FL. We find it quite robust in the low-frequency communication regime where FL is commonly believed to degrade drastically. We hope that our study could serve as a valuable reference for future practical usage and theoretical analysis in FL.

## 1 INTRODUCTION

Federated learning (FL) is a decentralized optimization framework in which several clients collaborate, usually through a server, to achieve a common learning goal without exchanging their data (Kairouz et al., 2019). FL has attracted a lot of attention lately due to the increasing concern about data privacy, protection, and ownership. The key challenge in FL is how to obtain a machine learning model whose performance is as good as if it were trained in a conventional centralized setting.

For models that are normally trained via stochastic gradient descent (SGD), such as deep neural networks (DNNs), FEDAVG (McMahan et al., 2017) is arguably the most widely used training algorithm in an FL setting. FEDAVG iterates between two steps: parallel local SGD at the clients, and global model aggregation at the server. In the extreme case where global aggregation takes place after every local SGD step, FEDAVG is very much equivalent to centralized SGD for training simple DNN models like multi-layer perception (Zhou & Cong, 2017; Stich, 2019; Haddadpour & Mahdavi, 2019; Li et al., 2020c; Zhao et al., 2018). Of course, due to the communication costs in practice, it is unlikely for clients to communicate at such a high frequency. Many existing works have thus focused on how to train DNNs at a lower communication frequency (e.g., once after local SGD for a few epochs), especially under the challenging condition where the data across clients are non-IID (Li et al., 2020b; Karimireddy et al., 2020; Acar et al., 2021; Chen & Chao, 2021).

In this paper, we specifically focus on DNN models that contain Batch Normalization (BN) layers (Ioffe & Szegedy, 2015). In centralized training, especially for deep feed-forward models like ResNet (He et al., 2016), BN has been widely used to improve the stability of training and speed up convergence. In the literature on FL, however, many of the previous experiments have focused on shallow ConvNets (CNN) without BN; only a few works have particularly studied the usage of BN in FL (Hsieh et al., 2020; Du et al., 2022). In Hsieh et al. (2020), the authors pointed out the mismatch between the feature statistics (i.e., the means and variances in BN) estimated on non-IID local data (during training) and global data (during testing), and argued that this cannot be addressed by using larger mini-batch sizes or other sampling strategies. Hsieh et al. (2020) thus proposed to replace BN with Group Normalization (GN) (Wu & He, 2018) and showed its superior performance in some extreme non-IID settings. Such a solution has since been followed by a long non-exhaustive list of later works (Jin et al., 2022; Charles et al., 2021; Lin et al., 2020; Yuan et al., 2021; Reddi et al., 2020; Hyeon-Woo et al., 2021; Yu et al., 2021; Hosseini et al., 2021).

With that being said, replacing BN with GN in FL is more like an ad hoc solution rather than a cure-all. First, in centralized training, BN typically outperforms GN empirically. Replacing BN with GN in FL thus seems like a compromise. Second, several recent works (Mohamad et al., 2022; Tenison et al., 2022; Yang et al., 2022; Chen & Chao, 2021) have reported that BN is still better than GN in their specific FL settings. Third, changing the normalization layer may create a barrier between the communities of centralized learning and FL. To illustrate, in centralized training, many publicly available pre-trained checkpoints (PyTorch, 2023; ONNX, 2023) are based on popular CNN architectures even recent transformers (Li et al., 2022) with BN; most understanding (Bjorck et al., 2018; Santurkar et al., 2018; Luo et al., 2019), empirical studies (Garbin et al., 2020), and theoretical analysis (Yang et al., 2019) about normalization in DNNs are built upon BN rather than GN. These prior results may become hard to be referred to in the FL community.

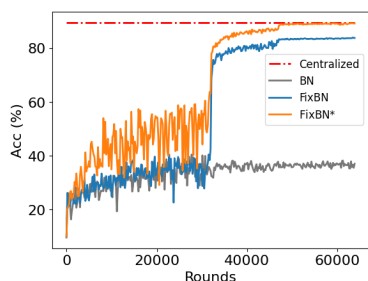

Figure 1: Our simple two-stage treatment FIXBN largely bridges the gaps of using BN in FL and centralized learning. Please see section 3 and section 5 for more details about FIXBN (⋆: with SGD momentum) and this non-IID CIFAR-10 experiments.

Last but not least, after a careful study of recent works that reported poor performance of BN (Wang et al., 2021; Zheng et al., 2020; Chai et al., 2021), we found that the huge gap between centralized learning and FL cannot be closed even if clients communicate right after *every* local SGD step. Such a finding sharply contradicts what is observed on DNNs without BN. In other words, the issue with applying BN in FL seems to be more fundamental than previously believed.

**Contributions.** Building upon these aspects, we strive to answer the following questions towards a more holistic understanding of BN in FL, especially under non-IID settings.

1. Why does BN degrade so drastically in FL compared to centralized training or other normalizers?

2. Is there a way to properly use BN in FL to bridge the performance gap w.r.t. centralized training?

3. Is there a comfort zone and danger zone for BN (and other normalization methods) in FL?

To begin with, we investigate several different perspectives to understand the issue of BN in FL, including BN statistic dynamics, the training/test mismatch of statistics, and the gradient w.r.t. the input of a BN layer under non-IID settings. Notably, we show that even if clients communicate after every local step, the *dependency of the gradient on the local mini-batch* prevents FEDAVG from recovering the gradient computed in the centralized training setting. We note that this does not happen to DNNs with GN, as GN does not use mini-batch statistics to normalize features.

Taking this insight into account, we propose a simple yet highly effective treatment named FIXBN, which requires no architecture change, no additional training, and no extra communication costs.

FIXBN follows the standard way to accumulate BN statistics during local training and aggregate them during global aggregation, just like FEDAVG. However, after a decent amount of communication rounds, FIXBN freezes the BN layer while keeping other DNN parameters learnable. In other words, in the second phase, local training does not rely on the mini-batch statistics but the frozen statistics to normalize features, which allows FEDAVG to *(a)* recover the centralized gradient under high communication frequency settings, *(b)* remove the mismatch of statistics in training and testing (Hsieh et al., 2020), and *(c)* retain the advantage of using BN statistics for normalization. As shown in Figure 1, FIXBN improves BN by a large margin in the high-frequency communication regime. FIXBN further applies to extensive FL tasks and settings, including common image classification benchmarks CIFAR-10 (Krizhevsky et al., 2009), Tiny-ImageNet (Le & Yang, 2015), and ImageNet (Deng et al., 2009a), and a natural non-IID semantic segmentation dataset Cityscapes (Cordts et al., 2015).

Along with our experiments comes a surprising observation of BN. Specifically, we find that BN is quite robust in low communication frequency or moderate non-IID settings, often outperforming GN. This not only (re-)suggests the use of cross-example statistics for normalization, but also suggests the need for a deeper investigation and theoretical analysis of BN in FL.

## 2 RELATED WORK

**Federated learning.** Many methods have been proposed to improve FEDAVG (McMahan et al., 2017) from different perspectives like server aggregation (Wang et al., 2020; Yurochkin et al., 2019; Lin et al., 2020; He et al., 2020; Zhou et al., 2020; Chen & Chao, 2021), server momentum (Hsu et al., 2019; Reddi et al., 2021), local training (Yuan & Ma, 2020; Liang et al., 2019; Li et al., 2019; Acar et al., 2021), pre-training (Chen et al., 2023; Nguyen et al., 2023), etc. Many works in FL have also gone beyond training a single global model. Personalized FL (Kairouz et al., 2019) aims to create models that are tailored to individual clients, i.e., to perform well on their data distributions. It is achieved through techniques such as federated multi-task learning (Li et al., 2020a; Smith et al., 2017), model interpolation (Mansour et al., 2020), and fine-tuning (Yu et al., 2020; Chen & Chao, 2022). In this paper, *we investigate the impact of the* BN *layers in DNNs in standard (not personalized) FL*.

**Normalization layers in centralized training.** Since BN was introduced (Ioffe & Szegedy, 2015), normalization layers become the cornerstone of DNN architectures. The benefits of BN have been extensively studied in centralized training such as less internal covariate shift (Ioffe & Szegedy, 2015), smoother optimization landscape (Santurkar et al., 2018), robustness to hyperparameters (Bjorck et al., 2018) and initialization (Yang et al., 2019), accelerating convergence (Ioffe & Szegedy, 2015), etc. The noise of the estimated statistics of BN in mini-batch training is considered a regularizer (Luo et al., 2019) that improves generalization (Ioffe & Szegedy, 2015). A recent study (Lubana et al., 2021) shows that BN is still the irreplaceable normalizer vs. a wide range of alternative choices in general settings. Note that, unlike in FL, BN *often outperforms* GN *in standard centralized training*.

**Existing use of normalizers in FL.** In the context of FL, less attention has been paid to normalization layers. Hsieh et al. (2020) is the first to suggest replacing BN with GN for non-IID decentralized learning. Several works (Du et al., 2022; Zhang et al., 2023) report that LN can be competitive to GN. Hong et al. (2021) enhances adversarial robustness by using statistics from reliable clients but not for improving performance. HETEROFL (Diao et al., 2020) proposed to simply normalize batch activations instead of tracking running statistics for the scenario that the clients have heterogeneous model architectures. These works aim to *replace* BN while *we analyze* BN *and reclaim its superiority*.

Several works (Duan et al., 2021; Idrissi et al., 2021) propose dedicated server aggregation methods for BN statistics (separated from other model parameters) for specific tasks. For multi-modal learning, Bernecker et al. (2022) proposes to maintain each modality as a different BN layer instead of sharing a single one. In personalized FL, Li et al. (2021); Andreux et al. (2020); Jiang et al. (2021) propose to maintain each client's independent BN layer, inspired by the practice of domain adaptation in centralized training (Li et al., 2016). Lu et al. (2022) leverages BN statistics to guide aggregation for personalization. The goals of these works are orthogonal to ours.

## 3 RETHINKING BATCH NORMALIZATION IN FL

### 3.1 BACKGROUND

**Batch Normalization (BN).** The BN layer is widely used as a building block in feed-forward DNNs. Given an input feature vector $\boldsymbol{h}$, the BN layer normalizes the feature (via the mean $\boldsymbol{\mu}_{\mathcal{B}}$ and variance $\boldsymbol{\sigma}_{\mathcal{B}}^2$ computed on a batch of features $\mathcal{B}$), followed by a learnable affine transformation (via $\boldsymbol{\gamma}, \boldsymbol{\beta}$):

$$\hat{\boldsymbol{h}} = f_{\text{BN}}(\boldsymbol{h}; (\boldsymbol{\gamma}, \boldsymbol{\beta}), (\boldsymbol{\mu}_{\mathcal{B}}, \boldsymbol{\sigma}_{\mathcal{B}}^2)) = \boldsymbol{\gamma} \frac{\boldsymbol{h} - \boldsymbol{\mu}_{\mathcal{B}}}{\sqrt{\boldsymbol{\sigma}_{\mathcal{B}}^2 + \epsilon}} + \boldsymbol{\beta}; \quad \epsilon \text{ is a small constant.} \tag{1}$$

In standard training, the statistics $\boldsymbol{\mu}_{\mathcal{B}}$ and $\boldsymbol{\sigma}_{\mathcal{B}}^2$ are computed on each training mini-batch during the forward passes. These mini-batch statistics are accumulated during training by the following exponential moving average (controlled by $\alpha$) to replace $\boldsymbol{\mu}_{\mathcal{B}}$ and $\boldsymbol{\sigma}_{\mathcal{B}}^2$ in Equation 1 for testing:

$$\boldsymbol{\mu} := \alpha \boldsymbol{\mu} + (1 - \alpha) \boldsymbol{\mu}_{\mathcal{B}}, \quad \boldsymbol{\sigma}^2 := \alpha \boldsymbol{\sigma}^2 + (1 - \alpha) \boldsymbol{\sigma}_{\mathcal{B}}^2. \tag{2}$$

**Federated Learning.** In a federated setting, the goal is to learn a model on the training data distributed among $M$ clients. Each has a training set $\mathcal{D}_m = \{(\boldsymbol{x}_i, y_i)\}_{i=1}^{|\mathcal{D}_m|}, \forall m \in [M]$, where $\boldsymbol{x}$ is the input (e.g., images) and $y$ is the true label. Let $\mathcal{D} = \cup_m \mathcal{D}_m$ be the aggregated training set from all clients;

$\ell$ is the loss function on a data sample. FL aims to minimize the empirical risk over all the clients:

$$\min_{\boldsymbol{\theta}} \ \mathcal{L}(\boldsymbol{\theta}) = \sum_{m=1}^{M} \frac{|\mathcal{D}_m|}{|\mathcal{D}|} \mathcal{L}_m(\boldsymbol{\theta}), \qquad \text{where} \quad \mathcal{L}_m(\boldsymbol{\theta}) = \frac{1}{|\mathcal{D}_m|} \sum_{i=1}^{|\mathcal{D}_m|} \ell(\boldsymbol{x}_i, y_i; \boldsymbol{\theta}). \tag{3}$$

The output $\boldsymbol{\theta}$ is the model parameters. For DNNs with BN layers, $\boldsymbol{\theta}$ includes learnable weights of the perceptron layers like fully-connected and convolutional layers, in addition to the statistics $\{(\boldsymbol{\gamma}, \boldsymbol{\beta}), \boldsymbol{S}\}$ of all BN layers, where $\boldsymbol{S} = (\boldsymbol{\mu}, \boldsymbol{\sigma}^2)$ are the BN means and variances. The $\boldsymbol{S}$, ideally, should approach the statistics estimated from the global data $\mathcal{D}$.

**Federated averaging (FEDAVG).** Equation 3 can not be solved directly in a federated learning (FL) setting due to the decentralized data. The fundamental FL algorithm FEDAVG (McMahan et al., 2017) solves Equation 3 by multiple rounds of parallel local updates at the clients and global model aggregation at the server. Given an initial model $\bar{\boldsymbol{\theta}}^{(0)}$, for round $t = 1, ..., T$, FEDAVG performs:

$$\textbf{Local: } \boldsymbol{\theta}_m^{(t)} = \texttt{ClientUpdate}(\mathcal{L}_m, \bar{\boldsymbol{\theta}}^{(t-1)}); \quad \textbf{Global: } \bar{\boldsymbol{\theta}}^{(t)} \leftarrow \sum_{m=1}^{M} \frac{|\mathcal{D}_m|}{|\mathcal{D}|} \boldsymbol{\theta}_m^{(t)}. \tag{4}$$

During local training, the clients update the model parameters received from the server, typically by minimizing each client's empirical risk $\mathcal{L}_m$ with several steps (denoted as $E$) of mini-batch SGD. For the locally accumulated means and variances in BN, they are updated by Equation 2. During global aggregation, all the parameters in the locally updated models $\{\boldsymbol{\theta}_m^{(t)}\}$, including the BN statistics, are averaged element-wise over clients. Typically, $E \gg 1$ due to communication constraints.

### 3.2 PROBLEM: BN IN FL CANNOT RECOVER CENTRALIZED PERFORMANCE

The main challenge in FEDAVG is that the distributions of local data can be drastically different across clients. This non-IID issue is particularly problematic for DNNs with BN layers since they depend on the activation mean and variance estimation computed on non-IID mini-batches.

We first consider communicating after every SGD step. That is, in the local training in Equation 4, we only perform a single mini-batch SGD update in each round, i.e., $E = 1$. At first glance, this should recover mini-batch SGD in centralized learning (e.g., training on multi-GPUs with local shuffling). However, as shown in Table 1 and Figure 1, even with high-frequency communication after every SGD step, there is a huge accuracy gap (about $45\%$) between centralized and federated learning for DNNs with BN. As a reference, *such a gap very much disappears for DNNs with* GN. Intrigued by this observation, we investigate the potential reasons from three aspects below, focusing on the non-IID FL setting.

Table 1: **FL with communication** *every step* ($E = 1$). We train a ResNet20 with either BN or GN on the non-IID CIFAR-10 dataset (5 clients, 2 classes per client). Both the FL and centralized training use SGD without momentum.

| Norm | Centralized Acc. | FL Acc. |
|---|---|---|
| GN | 87.46±0.57 | 87.37±1.16 |
| BN | 89.30±0.89 | 42.93±2.75 |

### 3.3 BN TRAINING DYNAMICS

We first consider the properties of BN in standard training. We note that BN normalizes the activations in the forward pass to ensure stable forward and backward propagation (Lubana et al., 2021). A naive workaround for the non-IID issue is to force all clients to normalize with the same statistics. We investigate this idea by "decoupling" the updates of the model weights and the BN statistics. Specifically, under the high-frequency communication setting with $E = 1$, we modify Equation 4 as follows. (a) At round $t$, given frozen weights in $\bar{\boldsymbol{\theta}}^{(t)}$, we update local statistics $\{\boldsymbol{S}_m^{(t+1)}\}_{m=1}^{M}$ via Equation 2 and aggregate them into $\bar{\boldsymbol{S}}^{(t+1)}$. (b) We then locally update the model weights in the evaluation mode, using the global statistics $\bar{\boldsymbol{S}}^{(t+1)}$ to normalize the activations. (c) Finally, we aggregate the local models into $\bar{\boldsymbol{\theta}}^{(t+1)}$. In the same FL experiment of subsection 3.2, we observe it achieves $52\%$ accuracy, still far from the BN centralized performance $89\%$.

We hypothesize the weights and statistics need to collaborate carefully to enjoy the benefits of BN dynamics. First, using fixed statistics in local training sacrifices the "sampling" noise of the estimated statistics from different mini-batch $\boldsymbol{S}_{\mathcal{B}} = (\boldsymbol{\mu}_{\mathcal{B}}, \boldsymbol{\sigma}_{\mathcal{B}}^2)$, which is believed to help search a flatter loss landscape (Luo et al., 2019). Second, using fixed statistics cannot properly normalize the activations in a mini-batch and could make DNN training fragile due to gradient explosion, especially in the earlier rounds of FEDAVG when the model weights and intermediate activations are changing rapidly.

### 3.4 RE-EXAMINING THE BN STATISTICS MISMATCH BETWEEN TRAINING AND TESTING

The reason why BN degrades in FL is believed to be the *statistics mismatch* issue pointed out by Hsieh et al. (2020). In section 5 of Hsieh et al. (2020), the authors argued that since the local accumulated statistics $\{\boldsymbol{S}_m = (\boldsymbol{\mu}_{\mathcal{D}_m}, \boldsymbol{\sigma}^2_{\mathcal{D}_m})\}$ are estimated on each of the non-IID local data $\{\mathcal{D}_m\}$, their average could be significantly different from the true statistics of the global data $\mathcal{D} = \cup_m \mathcal{D}_m$. In other words, the average of $\{\boldsymbol{S}_m\}$ (over $m$) may not be ideal in testing. To verify its impact on performance, we design a simple experiment (details in section 7) aiming to *remove* the statistics mismatch.

After the entire FEDAVG is finished, we re-accumulate the statistics $\{(\boldsymbol{\mu}, \boldsymbol{\sigma}^2)\}$ per BN layer directly on the aggregated training data $\mathcal{D} = \cup_m \mathcal{D}_m$ over clients, using Equation 2. Interestingly, we see a fairly small gain, i.e., less than $1\%$ accuracy gain even on extreme non-IID settings.

*Based on the verification, we surmise that while the statistics mismatch problem indeed has a minor impact, it seems unlikely to account for the primary performance drops of BN in FL.*

### 3.5 BN MAKES THE GRADIENTS BIASED IN LOCAL TRAINING

We hypothesize that under the non-IID settings, the major reason for the performance drop comes from BN's influence on local model training. As a simple illustration, we derive the forward-backward pass of the plain BN layer $f_{\text{BN}}$ (see Equation 1) for one example $\boldsymbol{x}_i$ in a mini-batch $\mathcal{B}$.

$$\textbf{Forward: } \ell(\hat{\boldsymbol{x}}_i) = \ell(f_{\text{BN}}(\boldsymbol{x}_i; (\boldsymbol{\gamma}, \boldsymbol{\beta}), (\boldsymbol{\mu}_{\mathcal{B}}, \boldsymbol{\sigma}^2_{\mathcal{B}}))) = \ell(\boldsymbol{\gamma}\frac{\boldsymbol{x}_i - \boldsymbol{\mu}_{\mathcal{B}}}{\sqrt{\boldsymbol{\sigma}^2_{\mathcal{B}} + \epsilon}} + \boldsymbol{\beta}) = \ell(\boldsymbol{\gamma}\tilde{\boldsymbol{x}}_i + \boldsymbol{\beta}); \quad (5)$$

$$\textbf{Backward: } \frac{\partial \ell}{\partial \boldsymbol{x}_i} = \frac{|\mathcal{B}|\frac{\partial \ell}{\partial \tilde{\boldsymbol{x}}_i} - \sum_{j=1}^{|\mathcal{B}|}\frac{\partial \ell}{\partial \tilde{\boldsymbol{x}}_j} - \tilde{\boldsymbol{x}}_i \cdot \sum_{j=1}^{|\mathcal{B}|}\frac{\partial \ell}{\partial \tilde{\boldsymbol{x}}_j} \cdot \tilde{\boldsymbol{x}}_j}{|\mathcal{B}|\sqrt{\boldsymbol{\sigma}^2_{\mathcal{B}} + \epsilon}}, \quad (6)$$

where $\ell$ is an arbitrary loss function on the BN layer's output $\hat{\boldsymbol{x}}_i$, "$\cdot$" is element-wise multiplication, and $\frac{\partial \ell}{\partial \tilde{\boldsymbol{x}}} = \boldsymbol{\gamma}\frac{\partial \ell}{\partial \hat{\boldsymbol{x}}}$. Please see Section 3 of Ioffe & Szegedy (2015) for the derivation of the gradient.

We can see that many terms in Equation 6 (colored in red) depend on the mini-batch features $\{\boldsymbol{x}_j\}_{j=1}^{|\mathcal{B}|}$ or statistics $(\boldsymbol{\mu}_{\mathcal{B}}, \boldsymbol{\sigma}^2_{\mathcal{B}})$. The background gradient $\frac{\partial \ell}{\partial \boldsymbol{x}_i}$ w.r.t. the input vector $\boldsymbol{x}_i$ is thus sensitive to what other examples in the mini-batch are. This is particularly problematic in FL on DNNs when clients' data are non-IID. Suppose $\boldsymbol{x}_i$ belongs to client $m$, the gradient $\frac{\partial \ell}{\partial \boldsymbol{x}_i}$ will be different when it is calculated locally with other data sampled only from $\mathcal{D}_m$ and when it is calculated globally (in centralized training) with other data sampled from $\mathcal{D} = \cup_m \mathcal{D}_m$. Such bias will propagate to the latter layers in a DNN. Namely, even if communicating after every mini-match SGD step, *how a particular data example influences the DNN parameters is already different between FL and centralized training.*

*We surmise that this is the fundamental reason why DNNs with BN degrade in FL. Although it becomes quite intuitive after our elaboration, to our surprise, such a gradient issue was not explicitly pointed out by previous works[1]. They mostly referred to the mismatch problem in (Hsieh et al., 2020).*

## 4 FIXBN: TOWARDS A PROPER USE OF BN IN FEDERATED LEARNING

### 4.1 ON *fixing* BN IN FL

Given the analysis in section 3, we ask, *Is there a way to bypass the issues of BN in FL to reclaim its superior performance in centralized training?* We start our exploration by taking a deeper look into the dynamics of BN statistics during standard FEDAVG training. Under the same $E = 1$ experiments in subsection 3.2, we highlight two critical observations from Figure 2 (details in the captions).

First, as shown in Figure 2 (a), the local mini-batch statistics remain largely different from the global statistics, even at later rounds, which results from the discrepancy between non-IID local data. This is not surprising. However, it re-emphasizes the potentially huge impact of the issue in subsection 3.5.

Second, still in Figure 2 (a), we look at each curve alone. We find that both the global and local mini-batch statistics essentially converge. We further show the variances of the local statistics within

---

[1]We recently noticed that a concurrent work (Wang et al., 2023) pointed out this finding as well. Nevertheless, our analysis and solution are quite different from theirs.

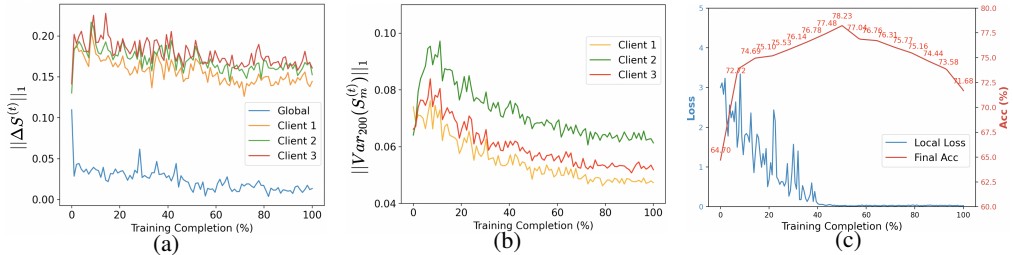

Figure 2: (a) Changes of global ($\|\bar{\boldsymbol{S}}^{(t+1)} - \bar{\boldsymbol{S}}^{(t)}\|_1$) and local mini-batch statistics ($\|\boldsymbol{S}_{m,\mathcal{B}}^{(t+1)} - \bar{\boldsymbol{S}}^{(t)}\|_1$). (b) Variances (running over $t - 200$ to $t$) of local statistics $\boldsymbol{S}_m^{(t)}$. (c) Loss of global model on the training data and **final-round accuracy** when freezing BN statistics at different intermediate rounds (CIFAR-10, $E = 100$).

each client become static in Figure 2 (b). This opens up the possibility to revisit the "decoupling" attempt in subsection 3.3.

Concretely, if local mini-batch statistics remain almost static in later rounds, replacing them with the fixed global statistics in local training may not degrade the benefits of BN. In contrast, it may fundamentally resolve the issue in subsection 3.5 — using the fixed global statistics in Equation 6 could prevent local gradients from diverging. We investigate this idea by replacing local mini-batch statistics with fixed global statistics starting at different rounds. As shown in Figure 2 (c), if the round is chosen properly, the *final accuracy* can be largely improved. Based on this insight, we propose our FIXBN method to address the issues in section 3.

### 4.2 TWO-STAGE TRAINING

To address the drawbacks discussed in section 3 simultaneously, we propose to divide FEDAVG training with BN into two stages, separated at round $T^\star$, inspired by the widely-used decay learning strategy for SGD (Robbins & Monro, 1951). Supported by the insight in subsection 4.1, we first follow standard FEDAVG to explore a decent model solution space, thanks to BN's important training dynamics as studied in subsection 3.3. Next, we propose to fine-tune the model for the rest of the training with the BN statistics *fixed*. This eliminates the statistics mismatch problem in subsection 3.4 since now training and test share the same BN statistics. It also addresses the biased gradients caused by non-IID statistics in subsection 3.5 as the local gradients no longer rely on mini-batches.

**Stage I: Exploration.** This stage is the standard FEDAVG with BN for two purposes: (a) to explore a proper model subspace without sacrificing BN's benefits on optimization (Ioffe & Szegedy, 2015); (b) to burn in the model and make it fitted to the training data. At the end, we save the aggregated statistics $\bar{\boldsymbol{S}}^{(T^\star)}$ of each BN layer from the average model $\bar{\boldsymbol{\theta}}^{(T^\star)}$.

**Stage II: Calibration.** We anticipate that the exploration stage already finds a proper region of the model solution, and calibrated fine-tuning is needed to further improve the performance. Starting from round $T^\star + 1$, we use the saved statistics as approximated global statistics to normalize the activations in local training. Since the model has been burned in, training with the fixed statistics is unlikely to suffer from the mentioned instability issue. In Figure 2 (c), we evaluate the training loss of the global model $\bar{\boldsymbol{\theta}}^{(T^\star)}$. It typically reaches a small loss after the first learning rate decay. Thus, *in experiments, we will simply fix the* BN *statistics since* 50% *of the total rounds of the FL training*.

While fairly simple, FIXBN cleverly leverages the global statistics to resolve the concerns in section 3, *with no architecture and objective change, no additional training and communication costs*.

## 5 MAINTAINING LOCAL SGD MOMENTUM

Besides BN, we identify another gap between FEDAVG and centralized training. While using SGD momentum in standard FEDAVG during local training is common, it will be discarded at the end of the round and re-initialized (along with any optimizer states) at the beginning of the next round of local training in FEDAVG. That is, the first several SGD steps in a round cannot benefit from it.

To further bridge the gap, we present a fairly simple method, which is to keep the **local momentum** without re-initialization after the end of the local training in each round. This makes it a stateful method suitable for cross-silo FL. Another stateless choice is to maintain **global momentum** (Karimireddy et al., 2020a) by uploading the local momentum to the server in every round and aggregating

Table 4: **Comparison to other FL normalizer methods.** Test accuracy (%) of ResNet20 on CIFAR-10 given # of rounds. The setting follows (Wang et al., 2023).

| FL Scheme | #R | IID | Non-IID |
|---|---|---|---|
| Centralized+BN | - | 91.53 | |
| Centralized+FixBN | - | 91.62 | |
| FEDTAN (Wang et al., 2023) | 580K | 91.26 | 87.66 |
| FEDAVG +BN | 10K | 91.35 | 45.96 |
| FEDAVG +GN | 10K | 91.26 | 82.66 |
| HETEROFL (Diao et al., 2020) | 10K | 91.21 | 30.62 |
| FEDDNA (Duan et al., 2021) | 10K | 91.42 | 76.01 |
| FEDAVG +FixBN (Ours) | 10K | 91.35 | 87.71 |

Table 5: ResNet20 with different normalization layers FL on CIFAR-10 (Shards, $E = 100$).

| Normalization Layer | Acc (%) |
|---|---|
| BN (Ioffe & Szegedy, 2015) | $53.97 \pm 4.18$ |
| GN (Wu & He, 2018) | $59.69 \pm 0.76$ |
| GN +WN (Qiao et al., 2019) | $66.90 \pm 0.81$ |
| LN (Ba et al., 2016) | $54.54 \pm 1.21$ |
| IN (Ulyanov et al., 2016) | $59.76 \pm 0.43$ |
| FIXUP (Zhang et al., 2019) | $70.66 \pm 0.24$ |
| **FixBN (Ours)** | $76.56 \pm 0.66$ |

it with Equation 4, for initializing the momentum of the next round of local training, with the cost of double message size. Empirically, we found the two methods yield similar gains (as will be shown in Figure 4) and recover centralized performance if communicating every step (Figure 1).

# 6 EXPERIMENTS (MORE IN THE APPENDIX)

## 6.1 MAIN RESULTS

We first compare our FIXBN (w/o momentum) against GN and BN on both class-non-IID and natural non-IID datasets. The group size of GN is tuned and set at 2. *The average of 3 runs of* FEDAVG *is reported.*

**Results on CIFAR-10 and Tiny-ImageNet.** We experiment on the standard FL benchmark CIFAR-10 (Krizhevsky et al., 2009) and Tiny-ImageNet (Le & Yang, 2015) with 5/10 clients, ResNet20/ResNet18 (He et al., 2016), respectively. For the hyperparameters, we generally follow (Hsieh et al., 2020) to use the SGD optimizer with 0.9 momentum, learning rate 0.02 (decay by 0.1 at 50% and 75% of the total rounds, respectively), batch size 20, and full participation, $E = 100$. We train **fixed 128 epochs** of total local SGD updates over all the clients and communication rounds. We consider **different non-IID degrees** including IID, Dirichlet(0.1, 0.3, 0.6) sampling follows (Hsu et al., 2019), and Shards that each client only has data for 20% of the classes. We show FIXBN consistently outperforms BN and GN in Figure 3, especially in severe non-IID cases. Surprisingly, we found BN can sometimes outperform GN. We provide a fine-grained comparison in section 7.

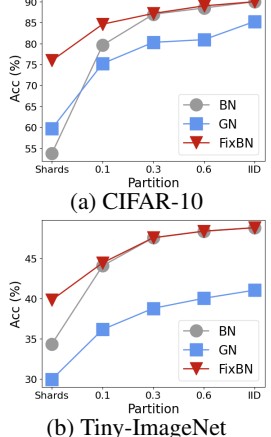

(a) CIFAR-10

(b) Tiny-ImageNet

Figure 3: Non-IID partitions with $E = 100$ steps.

**Results on ImageNet.** We extend FIXBN to ImageNet-1K (Deng et al., 2009b) dataset which is split into 100 clients Dirichlet (0.1) non-IID over classes. We learn a ResNet18 with 10% randomly sampled clients per round, 20 batch size, 0.1 learning rate (decay by 0.1 every 30% of the total rounds), 2 local epochs, and 64 epochs in total. Results in Table 2 show that FIXBN also perform the best.

Table 2: Class-non-IID ImageNet.

| Method | Network | Acc. $\Delta_{\text{-BN}}$ |
|---|---|---|
| GN | | $33.33 \pm 0.57$ |
| BN | ResNet18 (He et al., 2016) | $48.30 \pm 1.21$ |
| **FIXBN** | | $52.43 \pm 0.68$ (+4.1) |

Table 3: Pixel-wise accuracy and mean IoU (%) of image segmentation on Cityscapes .

| Method | Backbone | Mean IoU $\Delta_{\text{-BN}}$ |
|---|---|---|
| GN | | $43.2 \pm 0.33$ |
| BN | MobileNet-v2 (Sandler et al., 2018) | $48.9 \pm 0.36$ |
| **FIXBN** | | $54.0 \pm 0.29$ (+5.1) |
| GN | | $47.8 \pm 0.30$ |
| BN | ResNet18 (He et al., 2016) | $52.6 \pm 0.38$ |
| **FIXBN** | | $57.2 \pm 0.32$ (+4.6) |

**Comparison on realistic non-IID Cityscape.** We further consider a natural non-IID setting on the image segmentation dataset Cityscape (Cordts et al., 2015). We make each "city" a client and train 100 FEDAVG rounds using DeepLab-v3+ (Chen et al., 2018). More details are in the appendix. Results in Table 3 show that FIXBN's effectiveness is generalized to different architectures and vision tasks.

**Other FL baselines.** We reproduce Table 1 in FEDTAN (Wang et al., 2023) to compare to other BN variants FL methods in Table 4. We note that FEDTAN requires communication rounds linear to the numbers of BN layers $L$ as $\Theta(3L+1)$, which is much more expensive than FIXBN. HETEROFL (Diao et al., 2020) directly normalizes the activations, which cannot resolve the non-IID issue.

## 6.2 MORE ANALYSIS

**Maintained SGD momentum.** Next, we combine each normalizer with the maintained local momentum and global momentum proposed in section 5, respectively. We show FIXBN's effectiveness against BN and GN in Figure 4 in the (Shards, fixed epoch) setting with different numbers of local steps per communication $E$ of $\{1, 20, 100, 500, 2500\}$. We see FIXBN performs consistently better. More importantly, FIXBN remains highly accurate in fast communication, unlike BN, confirming that it mitigates the deviation issue in subsection 3.5 well. The improvements of using maintained global/local momentum are similar, providing the flexibility of stateless/stateful use cases. More gains are at small $E$, supporting our motivation to fix the zero initialization issue of the momentum. Across different settings, we see FIXBN $\geq$ BN $>$ GN in performance, consistent with Figure 3.

**Beyond BN & GN, is there any FL-friendly alternative?** So far, we mainly focus on BN. Our FIXBN does not require changing any architecture as for our motivation in section 1. Here we further compare to other normalization layers in Table 5. FIXBN still outperforms others. Interestingly, the normalization-free FIXUP (Zhang et al., 2019) initialization for residual networks[2] performs much better than GN, suggesting a new alternative in FL besides FIXBN.

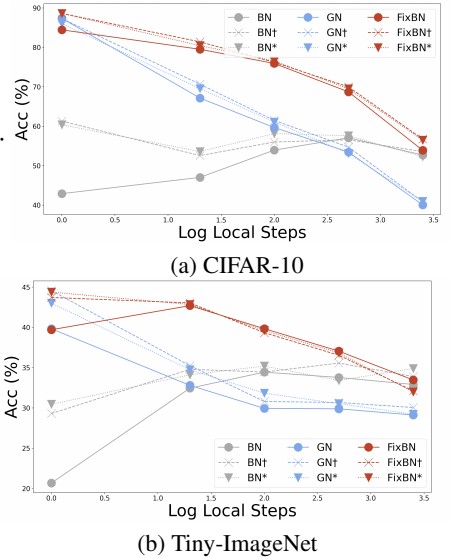

(a) CIFAR-10

(b) Tiny-ImageNet

Figure 4: **Maintained momentum.** Normalizers augmented with maintained **global momentum** (†) and **local momentum** (⋆) with different numbers of local steps per communication $E$.

## 7 A DETAILED STUDY OF BN VS. GN

Results in section 6 come to an unexpected finding: BN outperforms GN in many cases, contradicting the common belief that one should replace BN with GN in FL proposed in Hsieh et al. (2020) and followed by many works summarized in section 1 and section 2. To answer this question, we revisit the study in Hsieh et al. (2020) (which considers mere one FL setting) and provide a detailed study to compare BN and GN by varying several critical factors in FL to have a more complete picture.

**Experiment setup.** We focus on CIFAR-10 (Krizhevsky et al., 2009) and Tiny-ImageNet (Le & Yang, 2015) datasets, following the setup in section 6. We consider more factors like **(1) degrees of non-IID**, ordered in increasing skewness: **IID, Dirichlet(0.1, 0.3, 0.6), and Shards**. As practical FL is constrained on computation, we consider two **(2) budget criteria**: **fixed 128 epochs** of total local SGD updates over all the clients and communication rounds, and **fixed 128 rounds** of communication. In every round, each client runs $\{1, 20, 100, 500, 2500\}$ of **(3) local steps (E)**. We further include LeNet-like CNN (LeCun et al., 1998) for CIFAR-10.

### 7.1 RESVISITING: IS GN REALLY BETTER THAN BN?

We highlight the following observation from Figure 5, augmenting the findings in Hsieh et al. (2020):

- **No definite winners.** GN is often considered the default replacement for BN in previous FL works (section 1 and section 2). However, according to Figure 5, GN is not always better than BN.
- **BN often outperforms GN.** Instead, in most settings, BN outperforms GN. This can be seen from the green cells in "Acc(GN)-Acc(BN)" heatmaps of Figure 5.
- **GN outperforms BN merely in extreme cases.** We find that GN outperforms BN (the purple cells in "Acc(GN)-Acc(BN)" heatmaps) only in the extreme non-IID (e.g., Shards) and highly frequent communication (e.g., $E = 1$) settings. When clients cannot communicate frequently, the case where many existing FL works focus on, BN seems to be the better choice for normalization.

---

[2]Another concurrent work (Zhuang & Lyu, 2023) also reports improving by replacing BN with scaled weight normalization, similar to (Qiao et al., 2019) in Table 5.

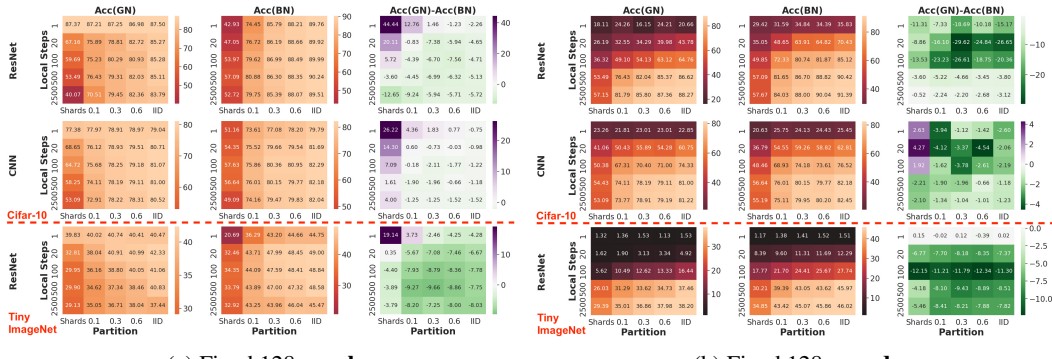

(a) Fixed 128 **epochs**  (b) Fixed 128 **rounds**

Figure 5: **Is GN always better than BN? No.** We compare their test accuracy in various FL settings on CIAFR-10 and Tiny-ImageNet, including different non-IID partitions and numbers of local steps $E$. Fixed budget of the total number of SGD updates (e.g., for CIFAR-10, 20 $E \times 5$ clients $\times 3200$ rounds $= 128$ epochs) or the number of total rounds (128 rounds) are given.

- **The trends along the number of local steps $E$ per communication round.** It is a perhaps well-known fact that increasing the number of local steps leads to greater drift as the local models become more biased (Karimireddy et al., 2020b). However, using more local steps also allows for more updates to the local models, potentially leading to an improved average model. To balance these competing considerations, we will discuss two criteria. For **(a) fixed epochs over all communication rounds**, a larger number of local steps means fewer communication rounds, in which GN degrades monotonically "as expected". *Interestingly,* BN *has an opposite trend.* BN actually improves and outperforms GN with larger $E$s. For **(b) fixed rounds**, understandably, using more local steps improves both GN and BN, since more local SGD updates are made in total. Nevertheless, the improvement saturates (e.g., $E \geq 500$).
- **Small difference from statistics mismatch.** In subsection 3.4, we discuss that the BN statistics mismatch problem might be minor. We re-estimate the statistics on global data and see a negligible accuracy gain from $44.09\%$ to $44.87\%$ on the Tiny-ImageNet (Dir(0.1), fixed epochs, $E = 100$).
- **More settings.** We verify in the appendix that factors like participation rates and the number of clients for partitioning the data do not change the above observation.

## 7.2 EFFECTS OF COMMUNICATION FREQUENCY

The constraint in communication, i.e., clients cannot aggregate the gradients frequently, is commonly believed as a major reason that hinders the performance of FL due to model drift (Karimireddy et al., 2020b). As BN cannot recover the centralized gradient even with high communication frequency and is outperformed by GN in such a setting, one may expect that BN will be consistently surpassed by GN when the frequency drops. But surprisingly, as observed in subsection 7.1, BN *is unreasonably effective when training with fewer communication rounds but more local steps per round.*

In Figure 6, we vary the number of local SGD steps per communication round (i.e., $E$) but fix the total number of SGD steps. We see the drastically different effect of $E$ on BN and GN. In particular, while the performance of GN drops along with increasing $E$, BN somehow benefits from a larger $E$. Such a discrepancy suggests the need for a deeper (theoretical) analysis of the usage of BN in FL.

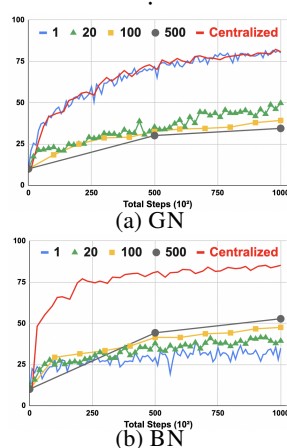

Figure 6: Test accuracy on CIFAR-10 for different local steps ($E$) per communication given a fixed number of SGD steps.

## 8 CONCLUSION

We revisit the use of BN layers and its common alternative, GN, in non-IID federated deep learning and conduct an in-depth analysis. We dissect the issues of BN in FL and propose a simple yet highly effective treatment named FIXBN to bridge the performance gap between FL and centralized training. We hope our study provides the community with a good foundation for the full (theoretical) understanding of the effectiveness of BN towards training deeper models in FL.

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
