# Appendix

We provide details omitted in the main paper.

- **Appendix A**: details of experimental setups (cf. section 7 and section 4 of the main paper).
- **Appendix B**: additional experimental results and analysis (cf. section 7 and section 4 of the main paper).

Table F: Summary of datasets and setups.

| Dataset | Task | #Class | #Training | #Test/Valid | #Clients | Resolution | Networks |
|---------|------|--------|-----------|-------------|----------|-----------|----------|
| CIFAR-10 | Classification | 10 | $50K$ | $10K$ | $5 \sim 100$ | $32^2$ | LeNet-CNN, ResNet-20 |
| Tiny-ImageNet | Classification | 200 | $100K$ | $10K$ | 10 | $64^2$ | ResNet-18 |
| ImageNet | Classification | 1,000 | $1,200K$ | $100K$ | 100 | $224^2$ | ResNet-18 |
| Cityscapes | Segmentation | 19 | $3K$ | $0.5K$ | 18 | $768^2$ | DeepLabv3 + {MobileNet-v2, ResNet-50} |

Table G: Default FL settings and training hyperparameters in the main paper.

| Dataset | Non-IID | Sampling | Optimizer | Learning rate | Batch size | $T^\star$ for FIXBN |
|---------|---------|----------|-----------|---------------|------------|---------------------|
| CIFAR-10 | Shards, Dirichlet($\{0.1, 0.3, 0.6\}$), IID | $10 \sim 100\%$ | SGD + 0.9 momentum | 0.2/0.02 | 20 | $50\%$ of total rounds |
| Tiny-ImageNet | Shards, Dirichlet($\{0.1, 0.3, 0.6\}$), IID | 50% | SGD + 0.9 momentum | 0.02 | 20 | $50\%$ of total rounds |
| ImageNet | Dirichlet 0.1 | 10% | SGD + 0.9 momentum | 0.1 | 20 | $50\%$ of total rounds |
| Cityscapes | Cities | 50% | Adam | 0.01/0.001 | 8 | 90th round |

## A  EXPERIMENT DETAILS

### A.1  DATASETS, FL SETTINGS, AND HYPERPARAMETERS

We use FEDAVG for our studies, with weight decay $1e{-}4$ for local training. Learning rates are decayed by 0.1 at $50\%, 75\%$ of the total rounds, respectively. Besides that, we summarize the training hyperparameters for each of the federated experiments included in the main paper in Table G. Additionally, for the Cityscape experiments in Table 3, we make each "city" a client and run 100 rounds, with local steps to be 5 epochs. More details about the datasets are provided in Table F.

For pre-processing, we generally follow the standard practice which normalizes the images and applies some augmentations. CIFAR-10 images are padded 2 pixels on each side, randomly flipped horizontally, and then randomly cropped back to $32 \times 32$. For Tiny-ImageNet, we simply randomly cropped to the desired sizes and flipped horizontally following the official PyTorch ImageNet training script. For the Cityscapes dataset, we use output stride 16. In training, the images are randomly cropped to $768 \times 768$ and resized to $2048 \times 1024$ in testing.

## B  ADDITIONAL EXPERIMENTAL RESULTS AND ANALYSIS

### B.1  ADDITIONAL STUDY OF FIXING BN PARAMETERS

In subsection 3.4, we discuss that the BN statistics are the main critical parameters in FL and thus motivate our design in FIXBN to fix the BN statistics to be the global aggregated ones after certain

rounds. Here we include a further study to confirm the importance of BN statistics by comparing them with the learnable affine transformation parameterized by $(\boldsymbol{\gamma}, \boldsymbol{\beta})$.

For FIXBN, besides fixing the BN statistics at round $T^\star$, we consider fixing the $(\boldsymbol{\gamma}, \boldsymbol{\beta})$ alone or together. The results on CIFAR-10 (Shards, fixed epochs, $E = 100$) setting using ResNet20 is in Table H. We observe that fixing the $(\boldsymbol{\gamma}, \boldsymbol{\beta})$ only has slight effects on the test accuracy either in combination with fixing $(\boldsymbol{\gamma}, \boldsymbol{\beta})$ or not, validating that the statistics are the main reason making it suffers more in FL, compared to the affine transformation. Fixing $(\boldsymbol{\gamma}, \boldsymbol{\beta})$ alone cannot match the performance of the originally proposed FIXBN.

Table H: **Fixing different parameters as in FIXBN.** We consider fixing the BN statistics $(\boldsymbol{\mu}, \boldsymbol{\sigma})$ as in original FIXBN or fixing the parameters $(\boldsymbol{\gamma}, \boldsymbol{\beta})$ of the affine transformation in BN layers. on CIFAR-10 (Shards, fixed epochs, $E = 100$) setting using ResNet20.

| $(\boldsymbol{\mu}, \boldsymbol{\sigma})$ | $(\boldsymbol{\gamma}, \boldsymbol{\beta})$ | Acc (%) |
|:---:|:---:|:---:|
| ✓ | ✓ | 75.22 |
| ✓ | ✗ | 76.56 |
| ✗ | ✓ | 55.33 |
| ✗ | ✗ | 53.97 |

## B.2 ADDITIONAL FIGURES FOR THE EMPIRICAL STUDY IN SUBSECTION 7.1

In subsection 7.2, we provide a detailed empirical study to compare BN and GN across various FL settings to understand their sweet spots. We provide a closer look at the observations we summarized in the main paper.

- **The trends along the number of local steps $E$ per communication round.** In subsection 7.2, we identify the opposite trends along #local steps $E$ between BN and GN. As shown in Figure G, we see GN drops with less communication as expected due to the well-known non-IID model drift problem in FL. Interestingly, we found that BN can actually improve within a certain range of communication frequencies (for local steps in [1,500]), which suggests that further investigation and theoretical analysis are required for BN in FL.

- **More settings.** We further verify that factors such as participation rate and the number of clients for partitioning the data in Figure H. As expected, the results are consistent with the observations summarized in subsection 7.1, particularly in that there is no definite winner between BN and GN while BN often outperforms GN.

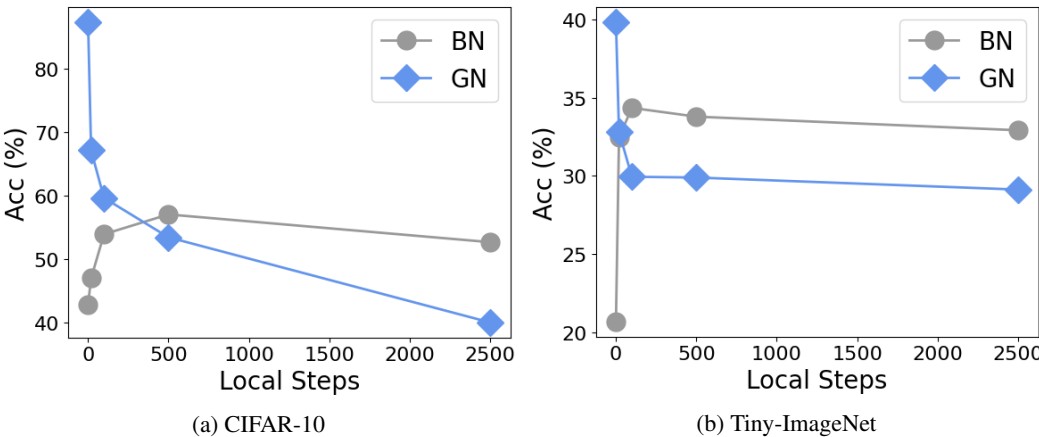

(a) CIFAR-10                              (b) Tiny-ImageNet

Figure G: **The opposite trends along #local steps $E$.** We consider the (Shards, **fixed epochs**) setting: the more the local step $E$ is, the fewer the total number of communication rounds is. GN drops with less communication as expected, while BN can improve.

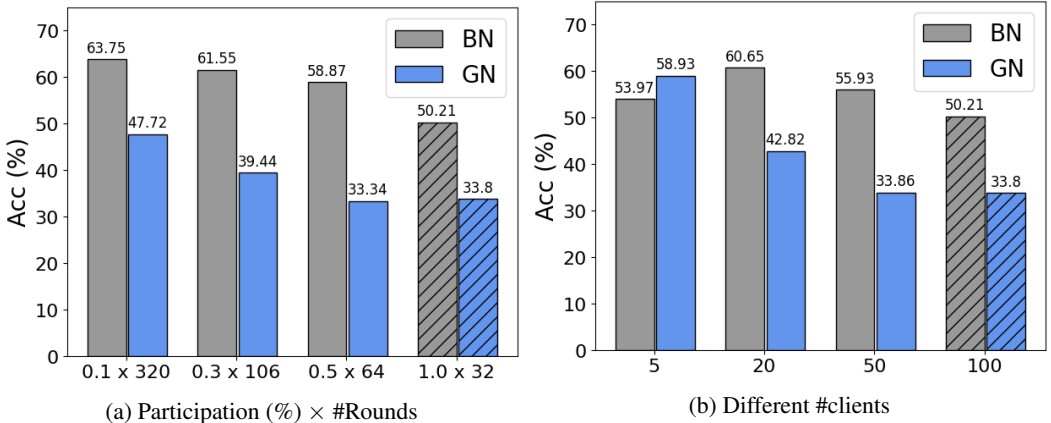

(a) Participation (%) × #Rounds

(b) Different #clients

Figure H: **More settings.** We consider more clients ($M = 5 \sim 100, E = 100$) for partitioning CIFAR-10 (Shards) with fixed epochs and varying the participation rate of clients every round.

### B.3 DIFFERENT # OF GROUPS FOR GN

For experiments in our study, we set the # of groups $= 2$ for GN layers. We did not find the group size a significant factor for the performance, as confirmed in Table I.

Table I: **Effects of the groupsize for GN.** We experiment with different # of groups ($2 \sim 8$) to divide the channels in GN layers in the CIFAR-10 (Shards, $E = 100$) with fixed epochs setting.

| Groupsize | Acc(%) |
|:---:|:---:|
| 2 | 59.42 |
| 4 | 57.61 |
| 8 | 58.86 |

### B.4 EFFECTS OF BATCH SIZE FOR BN

We experiment with various batch sizes for both BN and FIXBN in the CIFAR-10 (Shards, $E = 1$) setting and saw FIXBN maintains the advantage over standard FEDAVG +BN.

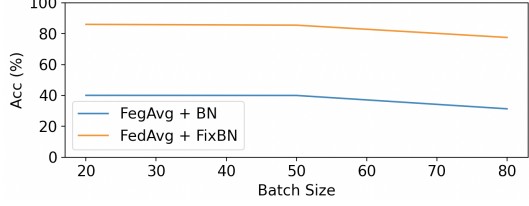

Figure I: FIXBN maintains advantage over different batch size selections.

### B.5 TRAINING CURVES

We provide the training curves of FIXBN and other normalizers under various settings in fixed 128 epochs using ResNet20 in Figure J, Figure K, Figure L, and Figure M, corresponding to section 7.

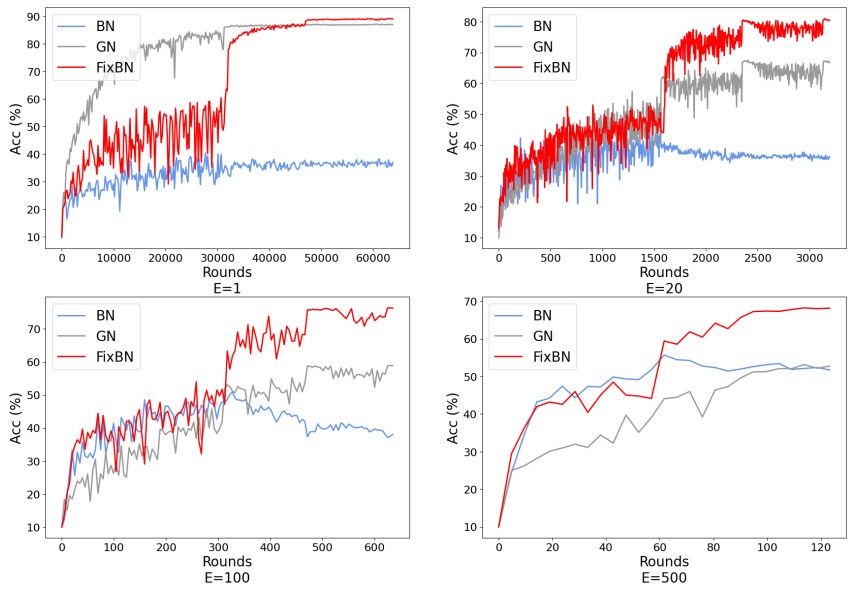

Figure J: Convergence curves of the test accuracy of CIFAR-10 with fixed epoch and **Shards non-IID** partitions, with $E = 1 \sim 500$.

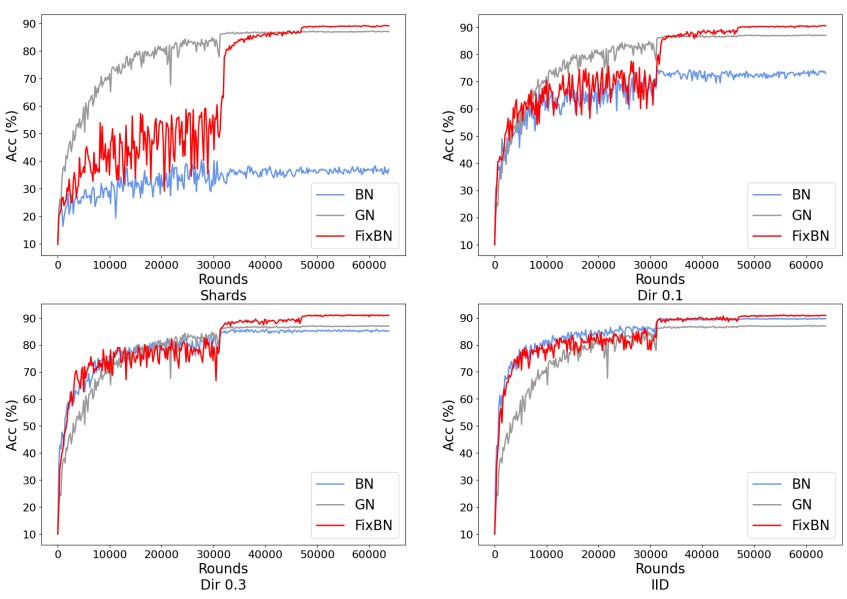

Figure K: Convergence curves of the test accuracy of CIFAR-10 in fixed epoch, **different non-IID partitions**, and $E = 1$ setting.

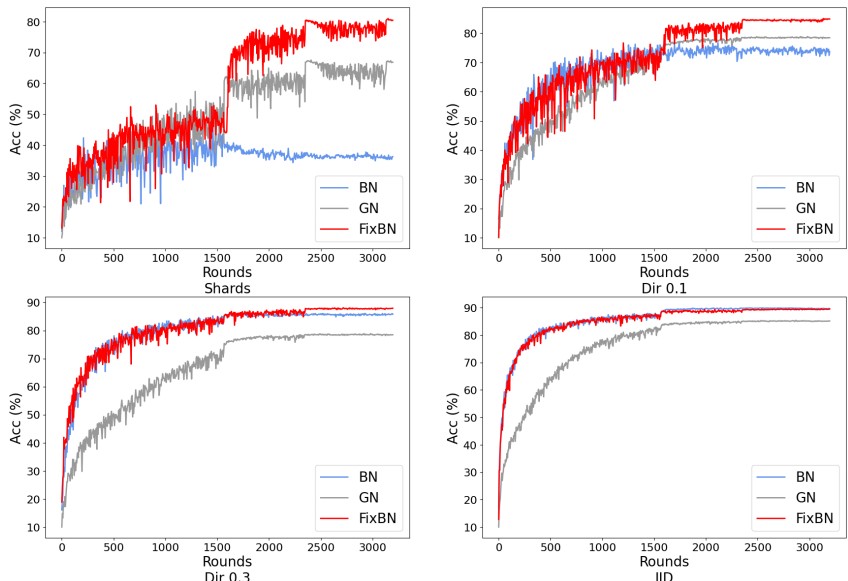

Figure L: Convergence curves of the test accuracy of CIFAR-10 in fixed epoch, **different non-IID partitions**, and $E = 20$ setting.

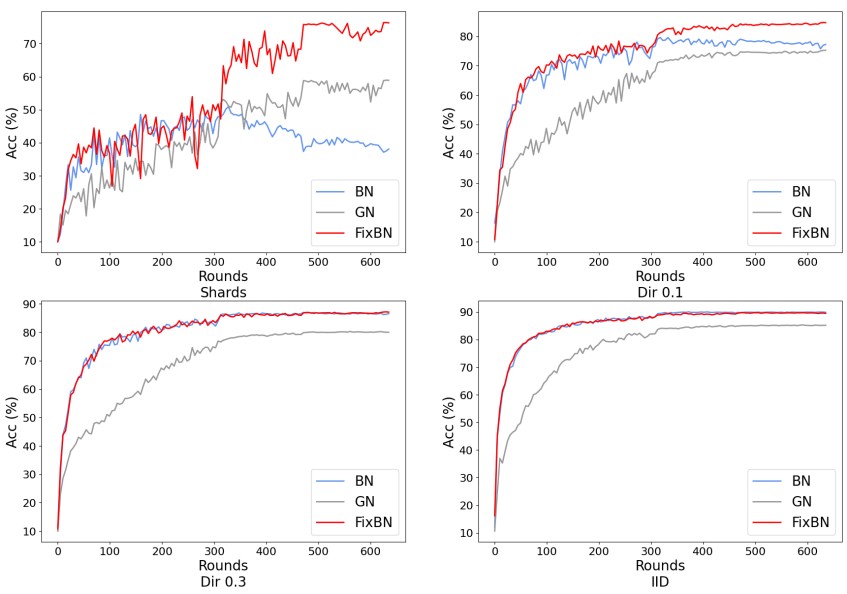

Figure M: Convergence curves of the test accuracy of CIFAR-10 in fixed epoch, **different non-IID partitions**, and $E = 100$ setting.

## C FIXBN ALGORITHM

---

**Algorithm 1:** FIXBN— federated learning with fixed batch statistics

---

**Server input** : initial global $\bar{w}_0$, fixing round $t^\star$

1 **for** $t \leftarrow 1$ **to** $T$ *rounds* **do**
2     **Communicate** $\bar{w}_t$ to all clients $m \in [M]$;
3     **for** *each client $m \in [M]$ in parallel* **do**
4        **if** $t > t^\star$ **then**
5           $\bar{w}_t \leftarrow \text{FixBNLayer}(\bar{w}_t)$;
6        **end**
7        $w_{t+1}^m \leftarrow \text{ClientUpdate}(m, \bar{w}_t)$; // `follow normal client update`
8        **Communicate** $w_{t+1}^m$ to the server;
9     **end**
10     **Construct** $\bar{w}_{t+1} = \frac{1}{M} \sum_{m=1}^{M} w_{t+1}^m$;
11 **end**

12 **FixBNLayer**$(w)$:
13     **for** *each BN module $w_{BN}$ in $w$* **do**
14        $w_{BN} \leftarrow \text{w}_{BN}.eval()$; // `global statistics will be used`
15     **end**

---