# OpenReview forum: "Making Batch Normalization Great in Federated Deep Learning"
_ICLR.cc/2024/Conference — Submitted to ICLR 2024_

### Official Review · Reviewer_4tfk · 2023-10-22

**Soundness:** 3 good
**Presentation:** 3 good
**Contribution:** 2 fair
**Rating:** 5
**Confidence:** 4

**Summary:**

This paper aims to understand the failure of Batch Norm in federated learning and proposes a new fix for it.

--post rebuttal--

Thanks for your additional experiment! I'm just wondering why is FixBN better than FedBN in the non-iid case, and which non-iid case are you experimenting on? From FedBN if it's covariate shift then FedBN should perform quite well.

**Strengths:**

1. Batch normalization is a common technique in deep learning but it fails drastically in FL. This work proposes a simple fix using fixed global statistics.
2. The paper is clearly written and studies important problems in FL.
3. The experimental section is quite comprehensive covering several image and segmentation datasets.

**Weaknesses:**

1. This paper lacks theoretical analysis for why this simple fix would work for BN in FL. It only proposes intuitive explanation.
2. Some references are missing. For example, [1, 2] study why BN fails in FL. [3] compares BN with GN and LN in with extensive experiments.
3. There is no comparison with FedBN (Li et al.), which also adapts BN to federated learning. Compared to all the previous works, I feel the contribution of this paper is limited.


*Minor: Sec 7.1 Resvisiting

Questions:
1. The conclusion in this work seems not to agree with [3], which says with Yogi and LN, FedAvg can approximate centralized training. What is the explanation for this?
2. Under covariate shift, is FixBN better than FedBN? And what happens under extreme label shift?


[1] Du, Z., Sun, J., Li, A., Chen, P.-Y., Zhang, J., Li, H., Chen, Y., et al. (2022). Rethinking normalization methods in
federated learning. arXiv preprint arXiv:2210.03277.
[2] Wang, Y., Shi, Q., and Chang, T.-H. (2023). Why batch normalization damage federated learning on non-iid data?
arXiv preprint arXiv:2301.02982.
[3] Guojun Zhang, Mahdi Beitollahi, Alex Bie, Xi Chen, Normalization Is All You Need: Understanding Layer-Normalized Federated Learning under Extreme Label Shift, arXiv:2308.09565.

**Questions:**

See above.

---

> ### Author Response · Authors · 2023-11-22
>
> Thank you for the references and we will make sure they are cited. Additionally, we thank the reviewer for their comments and address the weaknesses and questions below respectively.
> &nbsp;
>
> &nbsp;
>
>
> **W1**
>
> Although our study is primarily empirical, we respectfully do not think it’s just intuition. Rather, each step that leads to our proposed solution is supported by evidence from a carefully designed specific study.
>
> First, we note that table 1 reveals a crucial observation, the use of local batch statistics during local training degrades performance, as GN can recover centralized (-0% in accuracy) while BN degrades (-45%) when communicating at every step. This suggests that the use of local batch statistics in local training is what causes degradation. In other words, as long as local batch statistics are not used, the degradation can be rescued. This motivates us to design a local-statistics-free strategy in training.
>
> In sec. 4.1, we further examine this motivation. We examine both the BN statistics variance within local models and the gap of BN statistics between the local model and the global model. We found that while local statistics converge (fig 2b), the gap remains (fig 2a). The former suggests that if we replace the local statistics with global statistics after it has become static, then the benefits of BN will not be hindered (sec. 4.1 last paragraph); the latter essentially confirms that the local gradient issue will persist given that local gradient is highly dependent on local statistics in eq. 6 in sec. 3.5. Thus, it becomes clear that if we could choose a good moment to “fix” the BN statistics, we could not only enjoy BN benefits but also eliminate the degradation caused during local training.
>
> We can view our two-stage training separately where the first stage serves as a pre-training stage to get a good global statistics estimate and the second stage serves as BN-free training to rescue the degradation. With this regard the effectiveness of this simple “fix” is clearly understandable and the fact that FixBN perfectly recovers centralized learning when E=1 perfectly echoes the conclusion drawn from table 1.
>
> Additionally, we want to kindly share that developing theories is not the main goal of our paper. Our main contributions are to diagnose why and when BN fails in non-IID FL and provide a simple solution to reclaim the performance, as mentioned in the “Contributions” paragraph on page 2, sec. 1.
>
> &nbsp;
>
> **W2**
>
> We will make sure all 3 references are cited in the final version. For quick clarification, in addition to the external co-variate shift proposed by [A], we also identify local training as a source of failure of BN in FL. We also want to note that Wang et al., 2023 is a concurrent work and has already been addressed in the footnote on page 5 and compared with it in table 4 (FedTAN), in which our proposed method requires 50 times fewer communication rounds to achieve similar competitive performance. We address [B] in response to Q1 below.
>
> &nbsp;
>
> &nbsp;
>
> [A] Du, Z., Sun, J., Li, A., Chen, P.-Y., Zhang, J., Li, H., Chen, Y., et al. (2022). Rethinking normalization methods in federated learning. arXiv preprint arXiv:2210.03277.
>
>
> [B] Guojun Zhang, Mahdi Beitollahi, Alex Bie, Xi Chen, Normalization Is All You Need: Understanding Layer-Normalized Federated Learning under Extreme Label Shift, arXiv:2308.09565.

---

> ### Author Response · Authors · 2023-11-22
>
> **W3**
>
> Comparison with FedBN
> We agree with the reviewer that FedBN (Li et al.) is a well-known and important work in adapting BN to FL. We intentionally did not include it in our work because it is not a fair comparison. Specifically, we’d kindly point out that FedBN is proposed specifically for Personalized Federated Learning (PFL), which addresses a different learning objective than the Generic Federated Learning (GFL) setting that we consider (see eq. 3). We also explained both settings more explicitly in the “Federated Learning” paragraph in related work and made it clear that our focus is on GFL (last sentence in this paragraph). Nonetheless, we have conducted experiments with FedBN before and are happy to share them here. Following the same setting of table 4, we report the following results (in % accuracy):
>
> |    FL Scheme          |   IID       | non-IID
> | :----:  |    :----:   |          :----: |
> | FedBN  |   91.32   | 19.46
> |  Ours (FixBN)  |    91.35  |   87.71
>
> We note that FedBN allows each client to accumulate the local BN statistics without average during the FL process. In the above experiment, we simply average the clients’ BN statistics at the end of the FL process to construct a global model. Although the results show ours is 68.25% higher than FedBN, we still do not claim that our approach “outperforms” FedBN as, again, FedBN was proposed for a different purpose: personalized FL.
> &nbsp;
>
> &nbsp;
>
> **Contribution of our work**
>
>
> Please allow us to clarify. BN has been a long-standing issue in FL, and the widely accepted explanation in the community is by Hsieh et al., 2020 which attributes the cause to the mismatch between estimated global statistics and true statistics. Hsieh et al., 2020 also propose to replace BN with GN in all cases, a practice that many subsequent works have adopted. Although widely accepted, the mismatch issue was never explained by Hsieh et al., 2020. Moreover, our results show that even vanilla BN has better performance than GN in most cases (fig. 5), thus we believe bluntly replacing BN with GN in all cases is unfaithful.
>
> Given these motivations, our goal is to scrutinize the previous belief, diagnose why and when BN fails in non-IID FL, and provide a simple and clean solution to reclaim the performance, as mentioned in the “Contributions” paragraph on page 2, sec. 1. It is our deliberate intention to keep our solution as simple as possible.
>
> To the best of our knowledge, our work is the first of the kind that challenges the widely adopted results of Hsieh et al., 2020. We offer a more detailed analysis and evidence-based reasons of exactly why BN degrades in FL (sec 3) followed by a straightforward solution (sec 4) that could eliminate this issue in all cases (local SGD momentum: sec. 5 & sec. 6.2; different tasks: sec. 6.1; different communication frequency: sec. 7, etc.)
>
> To summarize, our study addresses a long-standing issue in FL. Its timeliness, simplicity, and strong performance could bring immediate benefits for the practitioners in the FL community and the industry to adopt and build upon. In addition, it also offers a good starting point for deeper theoretical study to follow.

---

> ### Author Response · Authors · 2023-11-22
>
> **Q1 The conclusion in this work seems not to agree with [3], which says with Yogi and LN, FedAvg can approximate centralized training. What is the explanation for this?**
>
>
> We thank the reviewer for raising the comparison here and are happy to provide an explanation. First of all, we respectfully think our work does NOT disagree with this work. In fact, we agree with this work that LN (we refer to it as GN later as it’s a more generic representation of batch-free normalization family and LN is basically GN when #of group = 1) can help bridge FL and centralized training (aka. approximate centralized training) when communicating each mini-batch step. As we can see, in fig. 4A, the leftmost blue dot represents FedAvg + GN at 1 mini-batch per communication, and we can see while centralized training is at 91.5% in accuracy, GN achieves 87.5%, which is already a good approximation. However, our work’s main focus is on BN and investigating and correcting BN’s failure in FL. In essence, they analyze why LN/GN can approximate centralized training while our intent is to ask, can we keep BN but also achieve the same goal? From our analysis and experiment it is clear that with correct treatment BN + FedAvg (FixBN) can indeed approximate centralized learning just as well if not better than GN + FedAvg. This conclusion means BN may just as well if not better approximate centralized learning in FL than GN but it is by no means a contradiction to the claim from [B].
>
> &nbsp;
>
>
> **Q2 Under covariate shift, is FixBN better than FedBN? And what happens under extreme label shift?**
>
> As mentioned in response to W3, we note again that FedBN is proposed for Personalized Federated Learning (PFL), which has a different learning objective than Generic Federated Learning (GFL) ours (eq. 3). We respectfully believe their results cannot be faithfully and fairly compared.
>
> We note that the results obtained in response to W3 are already conducted under the “shards” non-IID setting, where each client possesses some distinct classes. In this case, specifically, we have 5 clients each taking 2 classes for 10 total classes in the Cifar10 dataset. Under this condition, as we can see above, FixBN is 68.25% higher than FedBN in accuracy. However, we emphasize again here that we do not claim either one is “better” than the other per se due to the different objectives. Additionally, for the same reason, we believe comparing ours with it under covariate shift would not be meaningful either.
>
>
>
> &nbsp;
>
>
>
> [A] Du, Z., Sun, J., Li, A., Chen, P.-Y., Zhang, J., Li, H., Chen, Y., et al. (2022). Rethinking normalization methods in federated learning. arXiv preprint arXiv:2210.03277.
>
> [B] Guojun Zhang, Mahdi Beitollahi, Alex Bie, Xi Chen, Normalization Is All You Need: Understanding Layer-Normalized Federated Learning under Extreme Label Shift, arXiv:2308.09565.

---

### Official Review · Reviewer_6HP6 · 2023-11-01

**Soundness:** 2 fair
**Presentation:** 3 good
**Contribution:** 3 good
**Rating:** 5
**Confidence:** 4

**Summary:**

This work investigates the performance degradation of Federated Learning (FL) when coupled with Batch Normalization (BN). Contrary to the belief that this degradation results from mismatch in BN statistics between training and testing, the study claims that the problem lies in the sensitivity of backward gradients to the mini-batch composition. To address this issue, a two-stage training strategy is proposed, which involves fixing BN statistics after an initial normal training phase. This approach is shown to yield consistent improvements over Group Normalization (GN) across various FL scenarios.

**Strengths:**

- The two-stage BN training strategy introduced in this study is noteworthy for its simplicity, while proving to be highly effective in enhancing FL performance.

- The empirical results presented offer substantial support for the superiority of BN/Fixed BN over GN in a wide spectrum of FL setups.

**Weaknesses:**

- The study's exploration of the performance degradation in FL with BN remains primarily empirical. It would greatly benefit from a theoretical analysis of how FL aggregation influences the convergence of FL with BN, particularly in non-IID settings.
- The article falls short in providing a theoretical basis for the effectiveness of the two-stage BN training strategy in mitigating the degradation issue. A more in-depth examination of why the first stage's sensitivity to FL aggregation does not impede convergence and final performance would enhance the article's comprehensibility and credibility.

**Questions:**

The method's reliance on choosing when to fix the statistics during the remaining training is noted. However, the article does not address how this critical hyperparameter should be selected on the specific task.

---

> ### Author Response · Authors · 2023-11-22
>
> Thank you for recognizing the simplicity and effectiveness of our proposed approach. Here we address the weakness and questions below respectively.
>
> &nbsp;
>
> **W1**
>
> Although our study is indeed primarily empirical, we do not think it is trivial. Our finding that BN degradation happens due to local training but not global aggregation is drawn from careful analysis of two in-depth studies in sec 3.3 and sec 3.4. In sec 3.3, we investigate updating and aggregating model weights and BN statistics separately each round, which only achieves 51% accuracy when communicating each mini-batch step. In sec 3.4, we revisit the previously accepted belief that global aggregation drives global statistics different from true statistics (aka. the “mismatch” issue in Hsieh et al., 2020) and investigate rectifying the global statistics on a combined dataset which only yields a marginal improvement (<1%). Both these are still far from BN’s centralized performance at 89% in table 1, suggesting there are more fundamental reasons behind this. Hence comes our results on local training in sec. 3.5.
>
> We agree with the reviewer that more theoretical analysis could make the BN degradation issue more clear. We would like to politely and respectfully point out from our survey and consultation with experts in FL theory that, due to the non-linear optimization nature of DNNs, theory on BN in such networks is quite intractable to study even in a centralized setting, let alone in FL setting. Even the community-wise adopted belief and solution by Hsieh et al., 2020 on statistics mismatch and replacing BN with GN (last sentence of 3rd paragraph in sec. 1) provides no theoretical explanation and even analysis at all. Our study is motivated by the fact that given this widely adopted belief that GN is better for FL than BN, we found BN is still better than GN in most cases (page 9, fig. 5).
>
> Given these, our goal is to scrutinize the previous belief, diagnose why and when BN fails in non-IID FL, and provide a simple solution to reclaim the performance, as mentioned in the “Contributions” paragraph on page 2, sec. 1. We respectfully believe that the timeliness, simplicity, and strong performance of FixBN could bring immediate benefits for the practitioners in FL community and the industry to adopt and build upon and that our empirical study already serves as a strong starting point for any theoretical analysis to follow which we are willing and planning to undertake.
> &nbsp;
>
> **W2**
>
> **The article falls short in providing a theoretical basis for the effectiveness of the two-stage BN training strategy in mitigating the degradation issue.**
>
> We respectfully believe our motivation and analysis for using our two-stage FixBN already plots a rather clear picture of its effectiveness in mitigating degradation issues. First, we note that table 1 reveals a crucial observation, the use of local batch statistics during local training degrades performance, as GN can recover centralized (-0% in accuracy) while BN degrades (-45%) when communicating at every step. This suggests that the use of local batch statistics in local training is what causes degradation. In other words, as long as local batch statistics are not used, the degradation can be rescued. This motivates us to design a local-statistics-free strategy in training.
>
> In sec. 4.1, we further examine this motivation. We examine both the BN statistics variance within local models and the gap of BN statistics between the local model and global model and found that while local statistics converge (fig. 2B), the gap remains (fig. 2A). The earlier suggests that if we replace the local statistics with global statistics after it has become static, then the benefits of BN will not be hindered (sec. 4.1 last paragraph); the latter essentially confirms that local gradient will be continuously affected given our analysis that BN statistics drives local gradient in eq. 6 in sec. 3.5. Thus, if we could choose a good moment to “fix” the BN statistics, we could not only enjoy BN benefits but also eliminate the degradation caused during local training.
>
> Therefore, we can view our two-stage training separately where the first stage serves as a pre-training stage to get a good global statistics estimate and the second stage serves as BN-free training to rescue the degradation. With this understanding, our two-stage training strategy method could mitigate the degradation issue, as analyzed above, and its performance also perfectly echoes the conclusion drawn from table 1.

---

> ### Author Response · Authors · 2023-11-22
>
> **W2 (Continued)**
>
> **Why the first stage's sensitivity to FL aggregation does not impede convergence and final performance**
>
> As mentioned in the response above, sec. 4.1 underlines that the use of local batch statistics during local training leading to gradient bias (sec 3.5) is the main issue, and once these local batch statistics are no longer used, convergence will no longer be hindered (table 1, GN vs BN in FL setting). While the first stage’s sensitivity to FL aggregation does impede convergence in non-IID scenarios, it does not affect the performance after local batch statistics are fixed in the second stage. We’d also point out that the duration of the first stage does have some effect on the final performance, as fig. 2C shows the first stage duration needs to be chosen appropriately to fully release the full potential of FixBN.
>
> &nbsp;
>
> **Q1 The method's reliance on choosing when to fix the statistics during the remaining training is noted. However, the article does not address how this critical hyperparameter should be selected on the specific task.**
>
> We thank the reviewer for raising this concern and we are happy to explain it and will for sure add it to our final version to improve clarity. The timing to fix the BN statistics should be when the global statistics become static. This is studied in sec 4.1. As illustrated in fig. 2, the effectiveness of FixBN is largely associated with local mini-batch statistics (all lines in fig. 2A) or local loss (blue line in fig. 2c). As mentioned in the last paragraph in sec 4.1, the final performance of FixBN can be largely improved if the round is chosen properly. While these observations do not give a definitive answer to exactly when the “fix” should begin, they lead to three natural approaches: 1) assuming the training budget allows, the user could observe local mini-batch statistics in train time and visually determine a good “fix” point for their specific tasks in which the local mini-batch statistics or loss converge. 2) Alternatively, one could adapt a sliding window and decide based on the rate of change of the indicators within this window. Specifically, define $W$ as size of sliding window, define $\theta$ as the FixBN threshold, compute $\frac{1}{W} \sum_{i=t-W+1}^{t} (\text{var}_i - \overline{\text{var}}_W)^2$ (var is calculated the same as in fig 2B), if this value falls under $\theta$, then we apply FixBN. With this setup, we recommend using a large $W$ and small $\theta$ to account for instability during training. 3) Additionally, just like the general approach to determine a hyperparameter, it could be done by cross-validation, or we can apply federated hyperparameter optimization proposed in [A].
>
> &nbsp;
>
>
> [A] Federated Hyperparameter Tuning: Challenges, Baselines, and Connections to Weight-Sharing

---

> > ### Comment · Reviewer_6HP6 · 2023-11-23
> >
> > Thank you for the detailed responses. I maintain my perspective that this work is a empriical finding, and its performance is sensitive to the hypermeter when to fix the statistics.  Therefore, I choose to retain my original rating.

---

### Official Review · Reviewer_mhQT · 2023-11-01

**Soundness:** 3 good
**Presentation:** 3 good
**Contribution:** 2 fair
**Rating:** 5
**Confidence:** 5

**Summary:**

In this paper, the authors try to investigate the reason why batch normalization cannot work well when integrated into training of Federated Learning. They find that the data heterogeneity would lead to unstable gradient estimation. A new batch normalization method named FixBN is proposed for federated learning. A series of experiments are conducted to evaluate the performance of the proposed method.

**Strengths:**

1.	The authors conduct extensive experiments to evaluate the performance of the proposed method.
2.	This paper is well written and easy to read.

**Weaknesses:**

1.	This identified reason that “data heterogeneity would lead to unstable gradient estimation in federated learning” is to be expected and is also not new to the community.
2.	The proposed method is too straightforward and lacks technical depth. To be precise, adopts the original BN in the early stage of training and then fixes the BN layers. The authors are recommended to submit this paper to some workshops instead of the main conference.

Therefore, the contribution of this paper is limited.

**Questions:**

Please refer to my comments on weaknesses.

---

> ### Author Response · Authors · 2023-11-22
>
> Thank you for the comments, we address the weaknesses below.
>
> &nbsp;
>
>
> **W1**
>
> We do not claim the identified issue of "data heterogeneity leading to unstable gradient estimation in federated learning" as a novel contribution as it is known in the community. However, our main points are 1) that BN will amplify the gradient deviation (sec. 3.5) and 2) to challenge the widely accepted narrative that the degradation in BN performance is solely due to the mismatch between global and local statistics (i.e., FedAvg aggregation results in an inaccurate estimate of global statistics) raised by Hsieh et al., 2020 with our comprehensive diagnosis. As we demonstrate in sec. 3.5, rectifying the global statistics on a combined dataset yields a marginal (< 1%) improvement. We instead attribute the cause to local training.
>
> &nbsp;
>
>
> **W2**
>
> We respectfully disagree a straightforward solution is a weakness. Our method is simple but solves the issue directly after an in-depth analysis of the cause of why BN fails in sec. 3 and the motivations for fixing BN statistics in sec. 4.1. As machine learning researchers, we believe in Occam's razor: ``More things should not be used than are necessary’’, even in the deep learning era. We respectfully believe that if a simple method built upon solid empirical observations, motivations, and insights could be effective, then we really should not seek an even more complicated method even though it will make our work look fancier (and more novel superficially).
>
> We emphasize our contribution to its timeliness, simplicity, and effectiveness, being also easily adaptable for practitioners in the FL community. For example, we’d like to acknowledge a concurrent work by Wang et al., 2023, which also addresses the BN issue, but compared to it, our proposed method requires 50 times fewer communication rounds to achieve competitive performance, as shown in table 4. Besides simplicity, FixBN is also robust and effective across various conditions, including different normalization techniques (table 5), degrees of non-IID (fig. 3), client participation levels (fig. H), and numbers of local steps (fig. H). The simplicity and effectiveness of our approach, along with its ease of integration into existing FL systems, make it a timely and valuable contribution to the community, particularly for practitioners.
>
> For full disclosure, we received almost the same reviews in our previous submissions to other conferences. We truly appreciate the reviewer’s feedback and we have tried our best to incorporate them into this submission. We hope the reviewer can kindly reconsider the updated paper and the significance of our method to the FL practitioners during the evaluation.

---

> > ### Comment · Reviewer_mhQT · 2023-11-23
> >
> > Dear Authors,
> >
> > Thanks for your response. After reading the rebuttal, I increased my score to 5. But I still think the contribution of this paper is limited as the proposed method looks like a simple trick. Moreover, no significant improvements are achieved in the revisions.  It may be more suitable to submit this work into some workshop.

---

### Official Review · Reviewer_jccC · 2023-11-07

**Soundness:** 3 good
**Presentation:** 3 good
**Contribution:** 2 fair
**Rating:** 6
**Confidence:** 4

**Summary:**

This work focuses on the popular batch normalization method in FL, especially when the data is non-IID. Authors investigate the case where frequent communication is made between servers and clients, and reveal the biased gradient issue in local training. Experiments on BN and GN are conducted to show the empirical evidence. In the meanwhile, authors proposed a two stage training method to improve the performance of BN in various FL problems. Further, it is pointed out that the issue of BN does not occur when communication frequency is low.

**Strengths:**

1. This paper studies an important phenomenon in FL. It is widely observed in literature that BN can degrade performance in FL setting. However, most of the works find fix to the issue through changing the normalization method, e.g group normalization. This work focuses on the original method and conducts a comprehensive investigation on the vanilla BN, which provides a deep insight to the issue itself.
2. The experiment results are convincing and encouraging. Authors focus on two aspects of BN in FL: Biased gradient and communication frequency. It is pointed out that BN fail to have reasonable performance only when communication between server and clients is frequent. The experiment directly verify the conclusion.

**Weaknesses:**

1.  Some recent work also notices BN can degrade the performance of FL as well, e.g, Wang, Yanmeng, Qingjiang Shi, and Tsung-Hui Chang. "Why batch normalization damage federated learning on non-iid data?." arXiv preprint arXiv:2301.02982 (2023). This work contains comprehensive theoretical analysis of issue of BN and emphasizes that there exists deviation of gradient. However, authors do not include theoretical comparison with this work. Authors also mention that biased gradient can cause the failure of performance in FL, which is similar to the above work. Theoretical analysis is required to validate the conclusion.
2. Table 2 is an extreme case where single step of mini-batch SGD is performed. It is not surprising that such kind of high variance update can degrade the performance in distributed setting. More general setting should be considered to illustrate the issue of BN.
3. It is strongly suggested that specific algorithm is written in this paper. It is difficult to refer to equations to see the details.
4. The fix method is not novel. It sounds like an early stop for the local BN statistics and replacement with aggregated statistics in the second stage.

**Questions:**

1. How many epochs are performed in stage 1 and stage 2 respectively in the experiments?

---

> ### Author Response · Authors · 2023-11-22
>
> Thank you for your constructive comments, we will address the weakness and question below respectively.
>
> &nbsp;
>
> **W1**
>
> We thank the reviewer for mentioning this reference. This is a concurrent work; we recognize its contribution and have already cited it in the footnote on page 5 and compared our method with it in table 4 (see the row of “FedTAN”). Although their work did theoretically indicate the gradient deviation problem similar to our sec. 3.5, we further provide a more comprehensive analysis to verify the true cause of the failure of BN in FL, including the BN training dynamics caused by the mini-batch statistics in sec. 3.3 and fig. 2, re-examining the statistics mismatch issue pointed out by Hsieh et al., 2020 in sec. 3.4, and provide more extensive experiments on the effects of local SGD momentum (sec. 5 & sec. 6.2), different tasks (sec. 6.1), communication frequency (sec. 7), etc.
>
> We also agree with the reviewer that more theoretical analysis could elevate our work, but we want to kindly share that developing theories is not the main goal of our paper. Our main contributions are to diagnose why and when BN fails in non-IID FL and provide a simple solution to reclaim the performance, as mentioned in the “Contributions” paragraph on page 2, sec. 1. Compared to FedTAN (Wang et al., 2023), our proposed method requires 50 times fewer communication rounds to achieve competitive performance, as shown in table 4.
>
> &nbsp;
>
> **W2**
>
> We thank the reviewer for raising this issue. We would like to clarify that we did provide extensive studies on different/more general settings and the observations are consistent. Specifically, in sec. 7 (GN vs BN), we consider different networks (CNN & ResNet), different datasets (Cifar100 & Tiny-ImageNet), different communication frequencies (ranging from single mini-batch to 2500 mini-batches per communication), and different non-IID distributions (from shards to IID).
>
> The reason we focus on the extreme case of communicating every single step in some experiments is to highlight the negative effects of BN in FL. As elaborated in the second paragraph and table 1  in sec. 3.2, if we could communicate fast at every single step of mini-batch SGD in local training, then theoretically speaking standard FedAvg should already be able to recover centralized learning as this essentially emulates a distributed SGD scenario (training on multi-GPUs with local shuffling). As stated on page 2 of our paper, such a finding sharply contradicts what is observed on DNNs without BN.
>
> &nbsp;
>
> **W3**
>
> We have added the algorithm in the updated version of the paper, please see “FixBN Algorithm” in sec. C in the appendix.
>
> &nbsp;
>
> **W4**
>
> We’d like to respectfully denote that the novelty of the method is not the primary strength we wish to present. As mentioned in the response to W1 above, we address a long-standing issue in the community with a simple but extremely effective solution -- not only reclaim BN performance across the board but also requires 50 times fewer communication rounds to achieve this competitive performance compared to concurrent work (Wang et al., 2023). In addition, as machine learning researchers, we believe in Occam's razor: ``More things should not be used than are necessary’’, even in the deep learning era. We respectfully believe that if a simple method built upon solid empirical observations, motivations, and insights could be effective, then we really should not seek an even more complicated method even though it will make our work look fancier (and more novel superficially). In this regard, we’d rather consider the simplicity and straightforwardness a strength of our work.
>
> &nbsp;
>
>
> **Q1. How many epochs are performed in stage 1 and stage 2 respectively in the experiments?**
>
> All experiments are conducted with each stage having 50% of the total epochs of the whole federated training (i.e., the “fix” is performed from halfway through training). This design choice is based on our observation that local batch statistics have likely converged at this point (see fig. 2B and loss curve fig. 2C, the curve flattens roughly halfway). We found it a safe and stable choice.

---

### Author Response · Authors · 2023-11-22
**General Response**

We thank the reviewers for their valuable feedback. We have answered each reviewer's specific questions respectively. Here, we want to raise a general discussion with all the reviewers and ACs.
&nbsp;

&nbsp;

**Main contributions and significance**


We would like to kindly clarify that our main contributions are 1) we identify an overlooked aspect (why BN degrades in FL) and a widely accepted but potentially unfairly taken practice (replacing BN with GN in all cases in FL, revealed by fig. 5 and third paragraph of introduction in our study) proposed by Hsieh et al., 2020, which has over 450 citations as of today. 2) We provide a detailed analysis of this overlooked aspect (sec. 3) and propose a very clean and simple solution (sec. 4) that reclaims the performance of BN and resolves the long-standing issue of BN in FL in all cases studied (local SGD momentum: sec. 5 & sec. 6.2; different tasks: sec. 6.1; different communication frequency: sec. 7, etc.). Additionally, we would like to emphasize that developing theories is not the main goal of this paper and we believe our study can serve as a valuable reference for future practical usage and theoretical analysis in FL.

&nbsp;


**Simplicity of our method**


We would like to emphasize that it’s our deliberate intention to keep our method as simple as possible when it could have been made more complicated. We do not want to make our work look fancier and more novel superficially. We have explored more complicated methods such as updating and aggregating model weights and BN statistics separately each round in sec 3.3 which does not deliver in nearly as good a result as FixBN. Our simple method can deliver overwhelmingly strong results and already performs the best of all. This makes it easily understandable and adaptable especially for practitioners in this community. Hence we respectfully believe that simplicity is part of the strength of our work, rather than a weakness.

---

### Meta-Review · Area_Chair_2XL3 · 2023-12-07

**Metareview:**

Summary:
This work focuses on the popular batch normalization method in FL, especially when the data is non-IID. The authors investigate the case where frequent communication is made between servers and clients and reveal the biased gradient issue in local training. Experiments on BN and GN are conducted to show the empirical evidence. Meanwhile, authors proposed a two stage training method to improve the performance of BN in various FL problems. Further, it is pointed out that the issue of BN does not occur when communication frequency is low.


Strengths:
+ This paper studies an important phenomenon in FL.  This work focuses on the original method and conducts a comprehensive investigation of the vanilla BN, which provides a deep insight into the issue itself.
+ The experimental results are somewhat convincing and encouraging (for some reviewers).

Weaknesses:
- Relation to prior work (see reviews).
- Some results might highlight extreme/cornercases. A more general setting should be considered to illustrate the issue of BN.
- The clarity/presentation of the paper can be improved.
- At parts, the novelty of the paper is considered not novel.
- Some claims in the paper are considered straightforward: e.g., “data heterogeneity would lead to unstable gradient estimation in federated learning” is to be expected and is also not new to the community.
- The proposed method lacks technical depth / theoretical analysis.
- There is no comparison with some prior work.

**Justification For Why Not Higher Score:**

See above

**Justification For Why Not Lower Score:**

N/A

---

### Decision · Program_Chairs · 2024-01-16

Reject